# Distributions of recorded pain in mental health records: a natural language processing based study

Jaya Chaturvedi ![ORCID],[1] Robert Stewart,[1,2] Mark Ashworth ![ORCID],[3] Angus Roberts[1]

¹Institute of Psychiatry, Psychology and Neuroscience, King's College, London, UK
²South London and Maudsley NHS Foundation Trust, London, UK
³School of Population Health & Environmental Sciences, King's College, London, UK

**Correspondence to**
Jaya Chaturvedi;
jaya.1.chaturvedi@kcl.ac.uk

## ABSTRACT

**Objective** The objective of this study is to determine demographic and diagnostic distributions of physical pain recorded in clinical notes of a mental health electronic health records database by using natural language processing and examine the overlap in recorded physical pain between primary and secondary care.

**Design, setting and participants** The data were extracted from an anonymised version of the electronic health records of a large secondary mental healthcare provider serving a catchment of 1.3 million residents in south London. These included patients under active referral, aged 18+ at the index date of 1 July 2018 and having at least one clinical document (≥30 characters) between 1 July 2017 and 1 July 2019. This cohort was compared with linked primary care records from one of the four local government areas.

**Outcome** The primary outcome of interest was the presence of recorded physical pain within the clinical notes of the patients, not including psychological or metaphorical pain.

**Results** A total of 27 211 patients were retrieved. Of these, 52% (14,202) had narrative text containing relevant mentions of physical pain. Older patients (OR 1.17, 95% CI 1.15 to 1.19), females (OR 1.42, 95% CI 1.35 to 1.49), Asians (OR 1.30, 95% CI 1.16 to 1.45) or black (OR 1.49, 95% CI 1.40 to 1.59) ethnicities, living in deprived neighbourhoods (OR 1.64, 95% CI 1.55 to 1.73) showed higher odds of recorded pain. Patients with severe mental illnesses were found to be less likely to report pain (OR 0.43, 95% CI 0.41 to 0.46, p<0.001). 17% of the cohort from secondary care also had records from primary care.

**Conclusion** The findings of this study show sociodemographic and diagnostic differences in recorded pain. Specifically, lower documentation across certain groups indicates the need for better screening protocols and training on recognising varied pain presentations. Additionally, targeting improved detection of pain for minority and disadvantaged groups by care providers can promote health equity.

## STRENGTHS AND LIMITATIONS OF THIS STUDY

⇒ This study uses natural language processing on clinical notes to access a large sample of information about pain. The identification of such information would not be feasible manually.
⇒ This is the first cross-sectional study to summarise and describe the distribution of recorded pain within the clinical notes of mental health records.
⇒ The inclusion of both secondary mental health and primary care records for the same patients allows comparison of pain documentation across different health services.
⇒ When patients show no recorded pain, the study does not differentiate between pain that was discussed but not recorded or pain that was not discussed.
⇒ The findings are not generalisable to the general population since this study only looks at patients receiving mental healthcare within a specific geographic catchment.

## INTRODUCTION
### Background rationale

Pain and its relationship with mental health are important research topics. Pain has imposed a significant burden on society in terms of medical care costs as well as lost productivity.[1][2] Pain is multifaceted, with physical, psychological, social and biological causes and consequences.[3][4] In this context, pain refers to any pain condition or symptom, acute or chronic. Mental health disorders also present a considerable and complex public health problem, being a leading cause of disability and accounting for 28% of the national disease burden in the UK.[5] Electronic health records (EHRs) for mental health are a significant source of information for studying the intersection between pain and mental health among those who receive specialist service input. EHRs open up the possibility of investigating how pain is recorded and its impact on clinical outcomes.

Severe mental illnesses (SMIs) include diagnoses of schizophrenia spectrum disorder, bipolar disorder or severe major depressive disorder[6], where functional and occupational activities are severely impaired due to associated debilitating psychological problems.[7] While several studies have looked at the relationship between pain and schizophrenia and bipolar disorders[8–11] and at other mental illnesses such as depression[12–16], the complex

and potentially bidirectional nature of this relationship requires further understanding. Painful conditions occurring as physical comorbidities alongside mental health issues can exacerbate both conditions, with each impacting the management of the other. The combination of pain and depression, for instance, affects mental, physical and social functioning.[12] Furthermore, patients with schizophrenia tend to underreport their pain.[9] Analysis of secondary data sources, such as EHR databases, might help by providing a fuller picture of the recorded clinical presentation of this group of patients; however, a prerequisite is that pain is adequately represented in the derived data.

Demographic features such as age, gender and ethnicity may affect how pain is perceived and experienced. Pain affects twice as many people over the age of 60 as it does younger individuals.[17] While pain is not a natural feature of the ageing process, many health conditions causing pain become more common with increasing age. Nonetheless, older patients often believe pain to be a normal aspect of ageing and might be hesitant to report it.[17] There have also been variations in the reported perception of pain by female and male patients, with female patients reporting experiencing more pain than male patients.[18 19] Research has also shown disparities in pain perception across different ethnicities, with individuals of black African ethnicity reporting greater pain than their white counterparts.[20]

Socioeconomic status (SES) plays a role in health and overall well-being, with deprivation associated with unfavourable health outcomes and increased mortality rates.[21] Patients with SMI already experience higher mortality rates than the general population, and this discrepancy is exacerbated by socioeconomic deprivation, primarily due to unequal access to good quality physical healthcare services.[22–25] Furthermore, patients with SMI continue to experience a decline in their SES over time, compounding its impact.[26] Given these well-established connections between lower SES and reduced access to care[27 28], the examination of potential SES-based differences in the documentation of physical symptoms such as pain is particularly relevant. As disadvantaged patients face barriers in the screening for comorbid conditions, this may manifest in lower rates of discussions and recording of pain symptoms.

Most patient information is recorded in unstructured clinical narratives within EHR databases[29], and pain is likely to be no different, with few, if any, structured checklists ascertaining its presence in routine clinical care. Natural language processing (NLP), a computational approach to understanding and analysing human language, is therefore potentially useful for extracting such pain information. NLP has been applied extensively to EHR data, including studies of SMI, such as antipsychotic polypharmacy in mental healthcare[30], multimorbidity in individuals with schizophrenia and bipolar disorders[31] and extracting symptoms of SMI.[32]

In addition to secondary care data, it is also useful to consider the recording of pain in primary care data.

Within the UK, primary care (general practice (GP)) is generally the first point of contact for patients.[33] Exploring the overlap of recorded pain between primary care GP services and secondary care mental health services could, therefore, provide a more comprehensive view of the patient's pain experiences, and any discrepancies could highlight gaps in care and communication across different parts of the healthcare system. As primary care physicians are often responsible for the initial pain assessment and referral to specialists if needed, documentation patterns in GP records versus mental health provider records may differ for those with psychiatric disorders. Comparing recorded pain rates across these care settings can reveal insights into the consistency of pain screening among this vulnerable population across the healthcare landscape. This study used GP records specifically for patients contained within a mental health secondary care database in order to explore documentation patterns in primary care for patients who were recorded in secondary care with a mental illness. The analysis of these GP records takes the documentation beyond specialist mental health settings and provides valuable insights into the larger healthcare experiences of those with mental health disorders. Additionally, examination of potential differences or overlaps between primary and secondary care for the same patient cohort enables important observations about the consistency of pain assessment across providers.

### Objectives

The objective of this study is to describe the distributions of recorded pain among mental health service users according to demographic factors such as age, gender and ethnicity, as well as neighbourhood deprivation levels and mental health diagnoses. This was achieved by examining recorded pain through the means of an NLP application within the clinical text of a mental health EHR database and further evaluating this by measuring the overlap between pain recorded in secondary and primary healthcare, enabled through data linkage between the two.

This research aims to address knowledge gaps regarding the documentation of pain among mental health populations. In particular, a clearer understanding of these patterns is essential given the exceptionally high rates of pain conditions comorbid with mental health disorders. This study will answer fundamental questions about the frequencies of documented pain discussions during mental health encounters, consistency in pain detection across primary versus secondary care settings and whether certain groups defined by gender, ethnicity or SES face greater gaps in pain inquiry documentation. By analysing rates and differences of recorded pain within mental health records using a population-based cohort, this study works towards addressing needs around appropriate pain identification as a routine component of comprehensive mental health treatment.

## METHODS

### Reporting

We use the RECORD[34] guidelines and checklist, an extension of the Strengthening the Reporting of Observational Studies in Epidemiology[35] guidelines, for reporting the results of this study.

### Setting

Data on recorded pain, which in this context refers to any mentions of physical pain within the clinical notes, such as 'complains of pain' and 'experiencing headaches', were obtained from the clinical text of a mental health EHR database, the Clinical Record Interactive Search (CRIS) resource. This contains a de-identified version of EHR data from The South London and Maudsley NHS Foundation Trust (SLaM), one of Europe's largest mental healthcare organisations[36], which serves a geographic catchment of around 1.3 million residents in four south London boroughs (Croydon, Lambeth, Lewisham and Southwark). CRIS contains about 30 million free text documents, averaging 90 documents per patient.[29]

Data were also obtained from a primary care database called Lambeth DataNet (LDN)[37], which accesses all GP records from GPs based in the London borough of Lambeth. Data linkages (at the patient level) are already in place between CRIS and LDN.[38]

### Patient and public involvement

Patient and public involvement (PPI) in research is an active collaboration between researchers and members of the public, where the latter actively participate in contributing to the research.[39] A PPI group with lived experiences of SMI and chronic pain was consulted as part of this research. The nature of the data available was described to the group, and they were asked about their priorities regarding what research questions they would like answered. In response to this, the group was unanimously interested in further study of the differences in pain experiences based on demographics and diagnoses, and this was the main motivation for the objective of the study described here.

### Participants

A cohort of patients was extracted from the CRIS database, comprising those who were active (ie, the secondary care hospital trust (SLaM) has accepted them as a referral) and aged 18+ on the index date of 1 July 2018 and whose record contained at least one document (≥30 characters) within a window of 1 July 2017 to 1 July 2019.

LDN extraction followed similar criteria for patients who were active on the index date, aged 18+ and contained pain diagnoses or medications from 1 July 2017 to 1 July 2019. Free-text information is unavailable within LDN, so no document criteria were required.

### Variables

#### Demographics

Age, gender and ethnicity variables were extracted from structured tables within the CRIS database. Individuals with missing ethnicity values were retained as a separate category (not stated or known). In this context, ethnicity encompasses both race and ethnicity but is referred to simply as ethnicity for the sake of simplicity.

#### Diagnosis

The primary diagnosis recorded closest to the index date of 1 July 2018 was extracted from the structured tables within the CRIS database. These are coded using International Classification of Diseases-10th Revision (ICD-10).[40] The diagnosis codes were categorised as SMI and non-SMI, where SMI includes ICD-10 codes F20-29 and F30-33.

#### Deprivation

The Index of Multiple Deprivation (IMD) decile measures from 2019[41] were extracted for information on neighbourhood deprivation for each patient based on their address at the time of the index date aggregated by Lower Super Output Area (LSOA)—a standard national administrative unit containing an average of 1500 residents. National Census data are used to calculate IMD scores for each LSOA. A lower IMD decile indicates higher deprivation levels. Individuals with missing IMD scores were retained in a separate category.

#### Recorded pain

Pain-related keywords generated from a lexicon of pain terms[42] were used to identify patients in the cohort who had mentions of physical pain (such as 'worsening back pain', 'suffers from headaches' and 'complains of pain') recorded in their clinical notes within the predetermined window. The lexicon contains terms, such as aching muscles, back pain, headache, myalgia, etc, and can be accessed online (https://docs.google.com/spreadsheets/d/1z-6619UBdvwWrB9Sz4b1rbjDzuslOGCpts2DNc0naCc/edit?usp=sharing). An NLP application (F1 score, ie, harmonic mean of precision and recall: 0.98) was used on the documents of these patients. The application was developed by fine-tuning an existing model (SapBERT[43]) with 5644 gold standard annotations (triple annotated by medical student annotators) from CRIS, with the intention of classifying sentences within documents as relevant or not, where relevant refers to a mention of physical pain affecting the patient, such as 'complained of pain', 'has muscle aches…' and not relevant refers to no or negated mentions, hypothetical mentions and metaphorical mentions of pain, such as '…defensive of painful feelings' and '…painful consequences of using alcohol'. Only relevant mentions were used in the results reported here. The application has been described in detail in Chaturvedi *et al*.[44]

As with all other UK research based on access to anonymised primary care records, LDN does not allow access to any free text clinical notes. For this reason, pain information can only be extracted from the structured fields of the records. Read codes[45] were used to identify patients

who had a pain diagnosis or were on any pain medications and treatments:

1. A pain medication code list developed as part of a project described in Ma *et al*[46] focused on analgesics (obtained from dm+d, a dictionary of medicines and devices[47]) used in the treatment of 35 long-term conditions. These 35 conditions were obtained from Barnett *et al*,[48] a cross-sectional study on multimorbidities in patients registered with 314 medical practices in Scotland as of March 2007.

2. A pain diagnosis and treatment code list was developed as part of a collaboration project with outcome-based healthcare, an organisation that provides a platform for the study of population health outcomes,[49] with the research described in Hafezparast *et al*.[50] Pain diagnosis codes included instances such as back pain, referred ear pain, arthritis and trigeminal neuralgia. Pain treatment codes included codes for referral to a pain clinic, being seen in a pain clinic and being under the care of a pain management specialist.

While these codes were developed for chronic pain, they are generic enough to be used for this research, as highlighted in the examples mentioned. These code lists are available on GitHub (https://github.com/jayachaturvedi/pain_in_mental_health).

### Anatomy related to recorded pain

Another NLP application was developed as part of this research to identify the anatomy mentioned in relation to pain. This was a sentence classifier that generated a binary output: 'mentioned' or 'not mentioned'. The application was trained on 4026 gold standard sentences about anatomy mentioned in relation to pain and performed with an F1 score of 0.94. These gold standard sentences were a subset of the sentences used to train the pain NLP application. This application was run on sentences labelled as relevant by the pain application. Once the sentences that contained mentions of body parts were identified, they were run through MedCATTrainer,[51] which used named entity recognition, a type of NLP task, to label entities within the text to identify the specific body parts mentioned within the text. The purpose of using MedCATTrainer was to link the identified body parts to unique identification numbers (SCTID) from SNOMED CT, a terminology of clinical terms. These SCTIDs were used to aggregate the mentioned body parts for ease of analysis. For example, foot, calf and knee mentions would be aggregated under 'lower limb'.

### Overlap between CRIS and LDN

To examine the overlap across primary (LDN) and secondary (CRIS) care, the patient IDs from the CRIS cohort (n=27211) were searched for matching records within the LDN database over the same window of 1 July 2017–1 July 2019. Variables were generated indicating the presence of the patients within LDN, along with variables indicating the presence of any codes for pain medication, diagnosis or treatment based on the predefined

lists described above. This allowed the identification of patients with documented pain experiences in both their mental health and primary care records for the aligned time period. The cross-referencing process enabled the comparison of recorded pain between the two systems at the patient level.

### Descriptive statistics

All analysis was conducted using STATA V.15.1 and the Python programming language (V.3.10.0).

Descriptive statistics were obtained for demographic, deprivation and diagnosis features and compared between the two groups—patients who had recorded pain (class 1) and those who did not (class 0)—within their clinical notes. $\chi^2$ tests and logistic regression were conducted between the two classes to obtain adjusted ORs. The frequency of body parts affected by pain and the overlap of recorded pain experiences between CRIS and LDN were also reported.

## RESULTS
### Data extraction

Based on the extraction criteria, 27211 patients were represented. Among these patients, 18188 had pain keywords mentioned in their documents. These documents were run through the NLP application to label them as relevant to pain (class 1) or not (class 0), resulting in 14202 patients who had relevant mentions of pain within their clinical notes (figure 1). Relevant mentions include instances such as 'complains of pain', 'experiencing headaches', 'worsening back pain' and 'has stomach cramps'. Mentions that were not relevant were instances such as negations ('denied pain…' and 'no complaint of pain'), mentions within forms ('…experiencing other physical symptoms? for example, chest pain'), misspelt words ('…pained and decorated the walls'), hypothetical mentions ('reduce risk of pressure sores' and 'fear that eating will cause throat pain') and metaphorical mentions ('life is too painful to carry on living' and 'pain will end when she repents').

### Cohort characteristics and pain mentions

Among the cohort of 27211 patients, the mean age of the cohort was 44 (IQR range 29–55, SD 17.5), with 50.3% female and 48.2% of white ethnicity. The majority of the cohort (72.2%) lived in more deprived areas (IMD score ≤5), and 67% received a non-SMI diagnosis. 66.8% of the patients (18188 patients and 174167 mentions within documents) contained pain keywords within their documents, and 52.1% of the cohort (14202 patients) contained relevant mentions of pain in their documents.

Records of 52.1% of the patients within the cohort contained relevant mentions of pain. Differences between the patients who showed recorded pain (class 1) in their clinical notes and those who did not (class 0) are shown in table 1. Class 0 includes patients who did not have any mention of pain in their documents, as well as patients

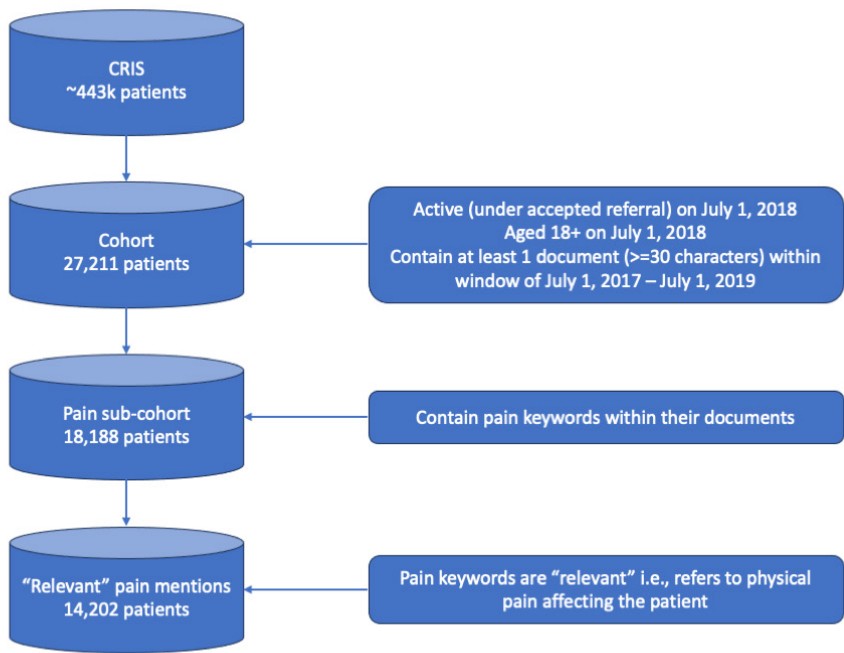

**Figure 1** Data extraction.

whose mentions of pain were classified as not relevant. Patients within class 1 had a mean of 10 pain mentions per document (median=4).

Demographic variations emerged between those with and without recorded pain in the cohort, as shown in table 1. The mean age was higher in patients with recorded pain at 46 (SD=17) compared with 41 (SD=17) for the remainder. Patients with recorded pain were more likely to be female and had a higher representation across all ethnic minorities. Additionally, patients with documented pain experiences were more likely to live in high-deprivation neighbourhoods.

Demographic, deprivation and diagnostic associations with recorded pain obtained through logistic regressions (unadjusted and adjusted for different factors as detailed below) are presented in table 2.

Unadjusted ORs revealed patients with documented pain experiences were more likely to be older (OR 1.17, 95% CI 1.15 to 1.19, p<0.001), females (OR 1.42, 95% CI 1.35 to 1.49, p<0.001), of Asian (OR 1.30 in relation to a white reference group, 95% CI 1.16 to 1.45, p<0.001) or black (OR 1.49, 95% CI 1.40 to 1.59, p<0.001) ethnicities and living in deprived neighbourhoods (OR 1.64, 95% CI 1.55 to 1.73, p<0.001) when compared with the respective reference groups. In a model containing all demographic variables (Model 1), the ORs for documented pain remained significant for all ethnic minority groups compared with the white group. With additional adjustment for neighbourhood deprivation (Model 2), the ORs were still significant for females relative to males. Similarly, in the model also adjusted for diagnoses (Model 3),

the ORs were also significant for females versus males. Patients with SMI had lower odds of documented pain (OR 0.43, 95% CI 0.41 to 0.46, p<0.001) than patients without SMI, which stayed significant after accounting for demographics, deprivation and diagnosis (Model 3). A supplementary model (Model 4) including both ethnicity and deprivation as covariates showed increased odds for Asian and black ethnicities when compared with white patients and those in more deprived neighbourhoods. The motivation for this model was to disentangle the independent contributions of ethnicity and deprivation to the differences in pain documentation. By adjusting for deprivation while evaluating the association between ethnicity and recorded pain (and vice versa), we can derive better effect estimates for each factor. This approach helps us to understand whether ethnicity-related differences persist after accounting for socioeconomic factors. We present selected incremental models for transparency in how estimates shifted with the inclusion of covariates but we focus our interpretation on the unadjusted and fully adjusted Model 3, which highlights the patterns with the most clarity.

### Anatomy distributions

Additional descriptive data were generated on the anatomical location of the pain reported. Among the 14 202 patients with any recorded pain, there were 174 167 mentions of pain within the documents. Of these, 7555 (53%) patients included 40 418 mentions of the anatomy associated with the pain. Of these 53%, each patient had an average of five body parts mentioned in the context of

**Table 1** Distributions between the two classes—class 0 (no recorded pain) and class 1 (recorded pain)

| Characteristics | N | Class 0 (no recorded pain) | Class 1 (recorded pain) |
|---|---|---|---|
| N (%) | 27 211 | 13 009 (47.9) | 14 202 (52.1) |
| Mean age (IQR) | 44 (29–55) | 41 (27–52) | 46 (32–56) |
| Gender (N, %) | | | |
| Male | 13 471 | 7037 (54.1) | 6434 (45.3) |
| Female | 13 709 | 5953 (45.7) | 7756 (54.6) |
| Not known | 31 | 19 (0.2) | 12 (0.1) |
| Ethnicity (N, %) | | | |
| White | 13 139 | 6014 (46.2) | 7125 (50.1) |
| Black | 5866 | 2115 (16.2) | 3751 (26.4) |
| Not stated/known | 4708 | 3418 (26.2) | 1290 (9.0) |
| Asian | 1506 | 592 (4.5) | 914 (6.4) |
| Other | 1197 | 512 (3.9) | 685 (4.8) |
| Mixed | 795 | 358 (2.7) | 437 (3.0) |
| Index of multiple deprivation (N, %) decile 2019 | | | |
| =5 more deprived) | 19 660 | 8847 (68.0) | 10 813 (76.1) |
| >5 (less deprived) | 6686 | 3836 (29.4) | 2850 (20.0) |
| Not known | 865 | 326 (2.5) | 539 (3.9) |
| Primary diagnosis: SMI vs non-SMI (international classification of diseases-9 code) (N, %) | | | |
| SMI | 8962 | 3059 (23.5) | 5903 (41.5) |
| Non-SMI | 18 249 | 9950 (76.5) | 8299 (58.5) |

SMI, severe mental illness.

pain. The most common body part affected by pain, as per the recorded mentions, was the lower limbs, which accounted for 20% of all mentions where anatomy could be ascertained (table 3).

Similar distributions were found within the SMI and non-SMI groups. Among patients with an SMI diagnosis, the most frequent body parts mentioned were the lower limbs (23%), upper body, excluding the back (17%) and stomach or abdomen region (15%). Patients with a non-SMI diagnosis most frequently reported lower limbs (19%), stomach or abdomen region (18%) and upper body, excluding back (17%), with minor variations in the frequencies.

### Overlap with primary care
When comparing secondary care CRIS records with those of primary care from LDN, among the 27 211 patients in the CRIS cohort, 4822 patients (17%) also had records in LDN. Among these patients who had records in both CRIS and LDN, 1507 (31%) patients were identified as having some recorded instance of pain in both their records, while 687 (14%) patients showed recorded pain only in LDN (primary care). Among the 27 211 patients within CRIS, 12 695 (46%) had recorded pain only within CRIS (mental healthcare), as seen in figure 2.

### DISCUSSION
This study investigated the differences observed in recorded pain mentions within the clinical notes of mental health records. The results reflect current literature findings that pain is a common issue among patients with mental health disorders. In a cohort of 27 211 patients, 18 188 (67%) patients contained pain-related keywords in their text, and 14 202 (52%) patients had relevant pain mentions, that is, the mention indicated physical pain affecting the patient in question. We found differences in documented pain mentions across genders, with a greater proportion recorded among female patients. This aligns with previous literature demonstrating gender differences in pain reporting and experiences.[16 47 48 52 53] Furthermore, while patients with known ethnicities had higher frequencies of recorded pain in the cohort of relevant pain mentions (in relation to those with unknown ethnicity), the most noticeable were the black, Asian and other ethnic groups. This highlights the need for a comprehensive exploration of pain experiences across diverse populations.[54] Moreover, the study's findings are also consistent with studies that indicate the impact of deprivation on health outcomes[21], as people living

**Table 2** Logistic regression findings for variables reflecting differences in class 0 (no recorded pain) and class 1 (recorded pain) (n=27 211)

| | Logistic regression models | | | | |
| | | Mutually adjusted | | | |
| | Unadjusted | Model 1 | Model 2 | Model 3 | Model 4 |
| --- | --- | --- | --- | --- | --- |
| Age (per 10 years) | 1.17 (1.15–1.19)* | 1.12 (1.11–1.14)* | 1.12 (1.11–1.14)* | 1.11 (1.10– 1.13)* | – |
| Gender | | | | | |
| Male | 1 (reference) | 1 (reference) | 1 (reference) | 1 (reference) | – |
| Female | 1.42 (1.35–1.49)* | 1.42 (1.35–1.49)* | 1.43 (1.36–1.50)* | 1.47 (1.40–1.55)* | – |
| Not known | 0.69 (0.33–1.42) | 1.08 (0.50–2.33) | 1.06 (0.49–2.30) | 1.10 (0.51–2.38) | – |
| Ethnicity | | | | | |
| White | 1 (reference) | 1 (reference) | 1 (reference) | 1 (reference) | 1 (reference) |
| Asian | 1.30 (1.16–1.45)* | 1.36 (1.22–1.52)* | 1.34 (1.19–1.49)* | 1.21 (1.08–1.36)* | 1.29 (1.15–1.44)* |
| Black | 1.49 (1.40–1.59)* | 1.58 (1.48– 1.69)* | 1.50 (1.40–1.60)* | 1.25 (1.17–1.34) | 1.42 (1.33–1.52)* |
| Other | 1.12 (1.00–1.27) | 1.20 (1.06–1.36) | 1.17 (1.03–1.32) | 1.10 (0.97–1.24) | 1.08 (0.96–1.33) |
| Mixed | 1.03 (0.89–1.18) | 1.15 (0.99–1.33) | 1.12 (0.96–1.30) | 1.06 (0.91–1.23) | 1.01 (0.87–1.17) |
| Not known | 0.31 (0.29–0.34)* | 0.36 (0.34–0.39)* | 0.37 (0.34–0.40)* | 0.40 (0.37–0.44)* | 0.32 (0.30–0.35)* |
| Index of multiple deprivation | | | | | |
| National decile ≤5 | 1.64 (1.55–1.73)* | – | 1.43 (1.35–1.51)* | 1.37 (1.29–1.45)* | 1.41 (1.33–1.50)* |
| Diagnosis | | | | | |
| SMI | 0.43 (0.41–0.46)* | – | – | 0.56 (0.53–0.59)* | – |

Values are given as OR (95% CI), and '*' indicates significance at p<0.001. Model 1 contained the demographic variables only (age, gender and ethnicity). Model 2 contained the variables from Model 1, plus the variable for deprivation (Index of Multiple Deprivation decile). Model 3 contained the variables from Model 2 plus the diagnosis variable. Model 4 contains the ethnicity and deprivation variables alone. Outcome is recorded pain versus no recorded pain.

in more deprived areas (IMD decile ≤5) were more frequently recorded with pain.

When comparing the overlap of patients between primary and secondary care, it was found that 17% of the patients within the CRIS cohort also had records within LDN. Among these patients, 31% had recorded pain instances in both records. While this overlap between primary and secondary care seems low, it is important to

**Table 3** Body parts affected (at mention level)

| Body parts | Mentions | Frequency (mention-level) |
| --- | --- | --- |
| Lower limbs | Feet, ankle, leg, knee, calf, thigh and toes | 20% |
| Upper body, excluding back | Chest, side of chest, upper body and torso | 19% |
| Upper limbs | Hand, wrist, arm, elbow, thumb and shoulder | 17% |
| Stomach/abdomen region | Stomach, abdomen, groin, bladder and prostate | 16% |
| Head and neck | Head, tooth, face, mouth, tongue, eye, ear and neck | 15% |
| Non-specific site | Entire body, skin, muscle and joint | 8% |
| Back | Back and lower back | 5% |

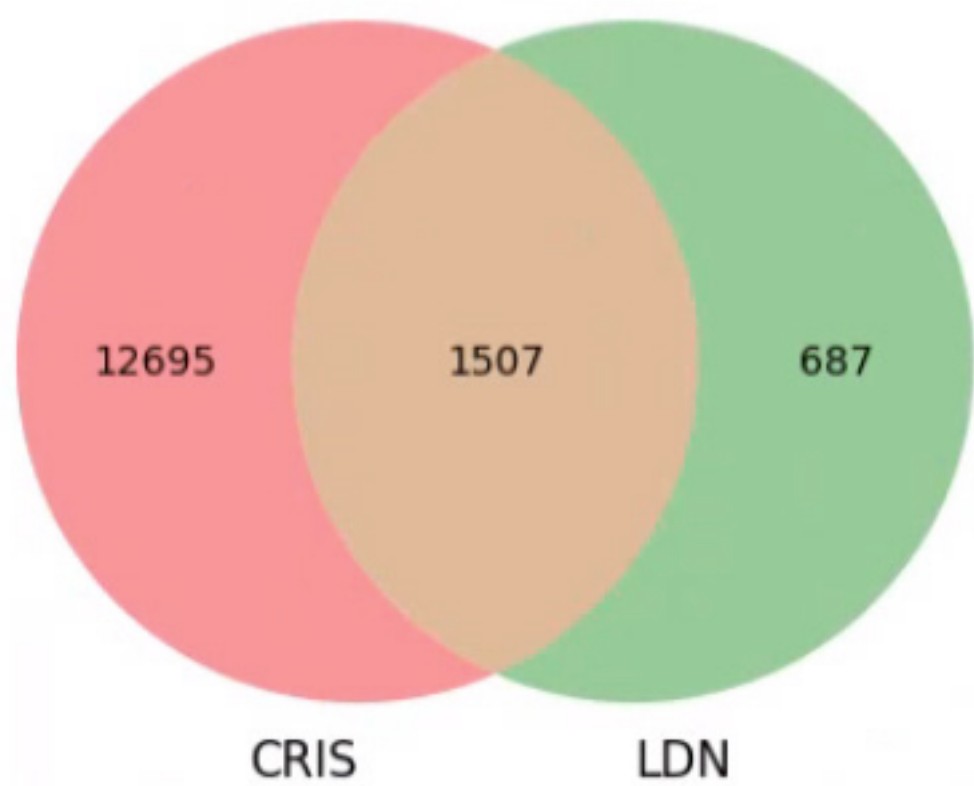

**Figure 2** Overlap of recorded pain in CRIS and LDN.CRIS, Clinical Record Interactive Search; LDN, Lambeth DataNet.

keep in mind that Lambeth only represents 22% of the catchment covered by CRIS.[55] Patients present in CRIS but not in LDN could include patients who have recorded instances of pain within the free-text clinical notes in LDN and might have been missed in this study since we do not have access to this text. Furthermore, this study did not differentiate between acute and chronic pain mentions and focused on extracting mentions of physical pain of any duration. As a result, the higher occurrence of pain mentioned within CRIS can be partially attributed to the documentation of such acute or short-lived pain episodes. Conversely, the GP records within LDN likely focus on recording persistent and chronic pain experiences. This disparity in recording pain should be considered when interpreting the findings of this study. Looking specifically at chronic pain instances within the CRIS notes may improve comparability. However, the temporal information required to determine pain chronicity from clinical notes is a particular challenge and can be difficult to extract reliably. Future work can attempt to differentiate acute and chronic pain through temporal or contextual information, which could provide richer insight. However, the current broad inclusion of pain provides wider coverage for this initial exploration of pain mentioned within clinical notes.

The findings of this study highlight important considerations that need to be made regarding the assessment and management of pain among people with SMI. Existing literature demonstrates that individuals with SMI often underreport pain symptoms yet experience disproportionately high rates of chronic pain conditions compared with the general population.[56 57] The low documentation of pain in the mental health records of this cohort indicates potential gaps in detection that warrant attention, particularly given research showing links between untreated pain and worse mental health outcomes. The challenges faced by this group in communicating their pain may inhibit pain identification.[58] Additional training focused on regular, thorough pain assessment within this group is needed for mental health professionals. Specifically, potentially implementing structured screening protocols, allowing patients to self-report through diverse modalities and increasing awareness of typical presentations could improve documentation and care standards. Pain assessment should become a routine aspect of comprehensive care for those with SMI to reduce compounding health decline. These steps towards more patient-centred pain management align with calls to better integrate physical and mental health services for this vulnerable population.

A strength of this study is the size of the data set available and the access to information about pain from the clinical text. To the best of our knowledge, this is potentially the first cross-sectional study to summarise and describe the distribution of recorded pain derived from routine mental health records. While the cohort data extraction did not apply any filters to demographics, aiming for

broad representativeness, other systemic biases related to access to healthcare resources may still exist. Factors like deprivation level and ethnicity can influence the utilisation of services and, therefore, documentation within health records, often stemming from perceived barriers to access. However, by not restricting cohort selection to demographic factors, this study intended to capture a diverse patient population receiving care across the South London boroughs.

A limitation of this study is its reliance on clinician documentation of pain within the clinical notes, which may be subject to a form of reporting bias. Specifically, the absence of documented pain could either be because patients were asked about their pain and this information wasn't recorded or because the patients were never asked about their pain. The absence of recorded pain does not indicate that the patients were not experiencing pain or that clinicians did not inquire about pain. This study methodology does not distinguish between these scenarios. The actual occurrences of pain experiences could remain unaccounted for if they weren't recorded by the clinicians or were not shared with them, especially for patients with severe mental illnesses who might be completely or partially non-verbal. While the NLP application achieved good performance metrics during its development and evaluation,[59] it is not impervious to imperfections. Instances of pain experiences might have been overlooked if they were not included as examples during the training of the application.

The scope of this study is limited to the examination of mental health records from an EHR database in South London. It is essential to acknowledge the potential influence of gender and ethnicity on the reporting of pain experiences, particularly if females and minority ethnicities (due to language barriers or other reasons) are less likely to self-report their pain experiences.[54 60 61] Since the focus of this study has been on a mental health EHR database, the clinical care within this setting is focused on mental health issues reported by the patients. Consequently, as much importance might not be given to the investigation and reporting of physical health conditions such as pain.

## CONCLUSION

The outcomes of this study have significant implications for the assessment and management of pain among patients with mental health disorders and highlight the importance of using NLP methods on EHR databases for research purposes. Notably, these findings reiterate the recommendations set forth by Mental Health America[62], advocating the need for proactive initiation of conversations around mental health and pain with patients. Relying solely on patients to self-report symptoms could potentially lead to worse outcomes, especially since the stigma surrounding persistent pain and mental health conditions may prevent patients from seeking the necessary treatment. Thus, early and proactive interventions could go a long way towards improved long-term outcomes. Unfortunately, there still exists a perceived lack of credibility and empathy towards patients living with pain[63], particularly when compounded by co-existing mental illnesses. This was one of the main points shared by the PPI group consulted as part of this study. More research in this area can help address these issues and provide safer and more equitable access to good-quality pain management.

It is possible that some patients within the cohort and, in general, within the CRIS database might be receiving psychological therapies for persistent pain. Future research leveraging the LDN-CRIS data linkage could examine referral patterns to these services. Examining referral trends over time and across demographic factors may uncover important insights regarding access barriers and ultimately enhance the delivery of appropriate psychological care for those suffering from pain. Analysing such patterns of psychological therapy referrals using the LDN-CRIS linked data can expand our understanding of this dimension of care.

While these findings represent a step forward, they are only one side of the story. Combining these findings with patient-reported insights could offer a more comprehensive understanding of pain experiences within this cohort. However, achieving this is a challenging task due to the lack of such data and the inability to link patient-reported experiences to their health records. Further research is needed to better understand the relationship between pain and mental health in this population. This could be achieved by accessing longitudinal data within this database and studying the temporal aspects of both conditions.

**Contributors** The idea was conceived by JC, AR and RS. JC conducted the analyses and drafted the manuscript. MA provided insights on LDN data. AR and RS provided guidance in the design and interpretation of results. All authors commented on drafts of the manuscript and approved the final version. JC, as guarantor, accepts full responsibility for the work and/or conduct of the study, had access to the data, and controlled the decision to publish.

**Funding** AR is funded by Health Data Research UK, an initiative funded by UK Research and Innovation, Department of Health and Social Care (England) and the devolved administrations, and leading medical research charities. RS is part-funded by (1) the National Institute for Health Research (NIHR) Biomedical Research Centre at South London and Maudsley NHS Foundation Trust and King's College London; (2) the National Institute for Health Research (NIHR) Applied Research Collaboration South London (NIHR ARC South London) at King's College Hospital NHS Foundation Trust; (3) the DATAMIND HDR UK Mental Health Data Hub (MRC grant MR/W014386) and (4) the UK Prevention Research Partnership (Violence, Health and Society; MR-VO49879/1), an initiative funded by UK Research and Innovation Councils, the Department of Health and Social Care (England) and the UK devolved administrations, and leading health research charities. JC is supported by the KCL-funded Centre for Doctoral Training (CDT) in Data-Driven Health. The funders were not involved in the study design, collection, analysis, interpretation of data, the writing of this article or the decision to submit it for publication. RS declared research support received in the last 36 months from Janssen, GSK and Takeda. All other authors declare no other competing interests. This work uses data provided by patients and collected by the NHS as part of their care and support. An application for access to the Clinical Record Interactive Search (CRIS) database for this project was submitted and approved by the CRIS Oversight Committee. The authors would like to acknowledge Dr Ruimin Ma for her help in obtaining the LDN codes.

**Competing interests** This paper represents independent research part-funded by the National Institute for Health Research (NIHR) Biomedical Research Centre at South London and Maudsley NHS Foundation Trust and King's College London. The views expressed are those of the authors and not necessarily those of the NHS, the NIHR or the Department of Health and Social Care. The funders had no role in study design, data collection and analysis, decision to publish or preparation of the manuscript.

**Patient and public involvement** Patients and/or the public were involved in the design, or conduct, or reporting, or dissemination plans of this research. Refer to the Methods section for further details.

**Patient consent for publication** Not applicable.

**Ethics approval** CRIS (as well as its associated linkages) has received ethical approval as a data resource for secondary analysis from the Oxford C Research Ethics Committee (reference 23/SC/0257). A patient-led oversight committee reviews and approves research projects that use the CRIS database. For service users, an opt-out system is in place and is advertised in all promotional materials and campaigns. Only authorised individuals can access this data from within a secure firewall. The CRIS project approval references for this work are 21-021 and 23-003.LDN approval was obtained as part of an existing CRIS project (project number 23-124) which included access to linked data from LDN (Caldicott Guardian approval, 15/9/21). This CRIS-LDN project aimed to examine the profile of patients with mental illnesses and chronic/persistent pain and compare them to controls from LDN who had chronic/persistent pain only.

**Provenance and peer review** Not commissioned; externally peer reviewed.

**Data availability statement** Data are available upon reasonable request. Data are owned by a third party, Maudsley Biomedical Research Centre (BRC) Clinical Records Interactive Search (CRIS) tool, which provides access to anonymised data derived from SLaM electronic medical records. These data can be accessed by permitted individuals from within a secure firewall (ie, the data cannot be sent elsewhere) in the same manner as the authors. For more information, please contact cris.administrator@slam.nhs.uk. Any STATA and Python code used in this project will be available on GitHub (https://github.com/jayachaturvedi/pain_in_mental_health).

**ORCID iDs**
Jaya Chaturvedi http://orcid.org/0000-0002-6359-9853
Mark Ashworth http://orcid.org/0000-0001-6514-9904

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
