## [Reviewer comments · BMJ Open]

ARTICLE DETAILS

TITLE (PROVISIONAL)	Distributions of Recorded Pain in Mental Health Records: A Natural Language Processing Based Study
AUTHORS	Chaturvedi, Jaya; Stewart, Robert; Ashworth, Mark; Roberts, Angus

VERSION 1 – REVIEW

REVIEWER	Henderson, Claire King's College London
REVIEW RETURNED	11-Oct-2023

GENERAL COMMENTS	This is on the whole a clearly written paper with a well considered limitations section. I have a few suggestions for improvement. Abstract CRIS and LDN are mentioned without either spelling out these acronyms or explaining what they are- if you lift text from results, rewrite for the abstract. Boroughs- for an international readership, I suggest local government areas. Introduction Black (African) Remove the brackets so that it is clear that what is meant is Black African as opposed to other Black groups Results Average of 10 pain mentions- suggest state either a mean or median depending on the distribution. Conclusion 'the stigma surrounding pain'- this applies chiefly to persistent pain so would be worth specifying. As with chronic fatigue, a small proportion of the patients in the CRIS dataset will be in receipt of psychological therapy to help manage persistent pain, with this as the reason for their having a CRIS record. So I wonder whether a future study using the linkage with the LDN could explore referral patterns for such therapy
--

REVIEWER	Kelly , Erin Thomas Jefferson University Hospitals
REVIEW RETURNED	06-Nov-2023

GENERAL COMMENTS	The authors have written about how well issues of pain are captured by mental health professionals within their electronic health record. This paper highlights how pain issues may be more or less prevalent or detected by clinicians. The authors cite some
--

	relevant literature but I think the framing of the paper and the discussion could be stronger if focused on how these results might inform clinical services and assessment. There is considerable literature on the relationship between pain and serious mental illness that could be better highlighted. It is well established that people with SMI are less likely to report pain than those without an SMI and the findings of this paper support that clinicians are less likely to record that within their mental health record, despite the high likelihood that they are experiencing pain. The implications for these findings for how they could shape clinical care and the need for additional training to ensure that clinicians are probing and recording about pain issues for those with SMI seems an important topic to explore further. The methods seem appropriate with the 80/10/10 approach and the sources of data used to create their dataset with one exception of the inclusion of the LDN dataset. It appears that they used a GP dataset that is not just limited to mental health records. A fuller justification for why they explored differences between the two datasets would be helpful. A little more detail about the NLP methods would also be helpful as much of that is explained in their prior papers but is not as well described in this paper. The use of logistic regression is appropriate, though the rationale for the inclusion of model 4 could be made stronger. Overall, there is considerable merit to this paper that could benefit from clearer rationale for why the study is important and for justification for the choices made.
--	--

REVIEWER	Meerwijk, Esther VA Palo Alto Health Care System, Center for Innovation to Implementation (Ci2i)
REVIEW RETURNED	21-Dec-2023

GENERAL COMMENTS	These are interesting data on pain recorded in mental health notes. The manuscript is organized and reasonably well written, but lacks specifics on a number of important aspects and requires clarification of additional aspects for the general reader. On the important aspects that are missing: 1) nowhere does it become clear what the problem is that is being addressed. The background says “the complex and potentially bidirectional nature of this relationship requires further understanding”, but why does it require further understanding? What don’t we know? 2) the conclusion claims that the results have significant implications for assessment and management of pain, but those implications are never mentioned. While the results are worth publishing because they show that physical pain is often reported in mental health notes, the study did not distinguish between ‘asked about pain and none reported’, ‘asked about pain but not recorded’, and ‘not asked about pain’, so to claim implications for assessment and management of pain seems premature. It would be helpful if the authors define what they mean by ‘recorded pain’ early on. I initially assumed it was pain intensity. Please add some example of pain terms. Abstract: Please avoid CRIS , LDN abbreviations as many readers will not know without explanation. Please specify the specific implications rather than saying they exist.
---

	Strength & Limitations: Please rephrase and look for strength and limitations of the methods. The current list, with exception of the last bullet, is neither strength nor limitation. Background: The second and third sentence are general statements but the citations to support them are specific to depression [1] and to HIV [2]. Please either add different literature that supports the general statements or rephrase. Demographic features don't "influence" pain perception. People with different features may experience pain differently, but the features themselves do not affect pain perception. Please rephrase. The paragraph about SES feels disconnected. Please drop or add why it is relevant to this study. Please clarify the distinction between primary and secondary care data, as this may not be clear for all readers. Methods: Under 'Participants', please clarify what "active (i.e., under accepted referral)" means. Under Recorded Pain: please add examples of pain terms so readers do not have to refer to the lexicon that is described elsewhere. Please also briefly describe the accuracy of the application that is used to classify sentences, so readers do not have to go somewhere else to find out about performance of the application. Please include some examples of 'metaphorical mentions of pain'. About codes for pain medications and diagnoses it says "While these codes were developed for chronic pain, they are generic enough to be used for this research" Please add how it was determined that they could be used for this study. The authors might want to consider mentioning this a potential limitation of the study. 'Another application' was used to determine anatomical location. Please briefly describe who developed the tool, how it works, as well as its accuracy in classifying body parts. Results: 18188 had pain key words but only 14202 relevant. Please add examples of pain words that were not relevant. It would make sense to combine Cohort characteristics and the Pain mentions section. For Table 2 please describe what is being predicted. Page 12, line 11: "compared to the remainder of the sample" is not correct. Given that these are regression models, the comparison is always with the group that makes up the reference, not the remainder of the sample. Related, on line 21 about 'independent increased odds', this is not correct. The comparison is Asian vs. White and Black vs. White. Not Asian vs. Black. Please rephrase. Odds ratios becoming stronger or strengthening is vague. Please rephrase. The authors probably intend to say that the ORs are higher, compared to another model, but since they are not comparing model fit we can not say whether higher is better, more accurate, or stronger etc. On the topic of not comparing model fit, what is the purpose of models 2 and 4? I suggest just reporting unadjusted and model 3.
--	---

	On anatomy distribution, please rephrase “the nature of the pain”. It suggests labels like ‘stinging’, ‘dull’, ‘throbbing’, but what is meant is anatomical location. It would be of interest to see the distribution of locations by SMI (yes/no). Discussion: Some language in the discussion is inaccurate. For example, females being overrepresented, gender disparities in pain experiences. More female patients had pain mentions. Overrepresented means that there were more female patients than should have been. ‘Disparities’ suggests some external factor that caused the difference in pain mentions. This study was purely descriptive and did not look into those factors. Please rephrase and avoid using those terms. The suggestion that the results align with undertreatment of minority patients should be removed. As there can be multiple reasons why some groups have higher pain mentions than other (see main concern 2 above) it is premature to suggest this is the result from undertreatment. A limitation of the study is not that it depends on physicians recording, but that you cannot distinguish whether asked and not recorded or not asked at all. The sentence “While the NLP application achieved good performance metrics during its development and evaluation” needs a citation. Generalizability of an exploratory and descriptive study like this is not the point of the study. I suggest dropping the part about generalizability. Please drop the last paragraph of the discussion. The study was descriptive so cause & effect is not at stake. Moreover, the study did not show disparities. It did show differences in recorded pain mentions but the cause of that was not addressed by this study and hence the authors cannot say that disparities exist. Conclusion: Please omit the last sentence as it cannot be concluded from this study. The study looked at pain mentions in clinical records, not at the relationship between pain and mental health or interventions to manage pain.
--	--

VERSION 1 – AUTHOR RESPONSE

Reviewer 1

Comment: Abstract - CRIS and LDN are mentioned without either spelling out these acronyms or explaining what they are.

Response: The abbreviations have been removed. Location: Abstract - Results - last 2 lines of the paragraph.

Comment: Boroughs- for an international readership, I suggest local government areas.

Response: Boroughs has been replaced with local government areas. Location: Abstract - Design, Settings and Participants - last line of the paragraph.

Comment: Introduction - Black (African) - Remove the brackets so that it is clear that what is meant is Black African as opposed to other Black groups

Response: Brackets have been removed. Location: Introduction - Background Rationale - end of 3rd paragraph.

Comment: Results - Average of 10 pain mentions- suggest state either a mean or median depending on the distribution.

Response: This has been changed to mean and included a median as well. Location: Results - Cohort Characteristics and Pain Mentions - end of second paragraph.

Comment: Conclusion - 'the stigma surrounding pain'- this applies chiefly to persistent pain so would be worth specifying.

Response: persistent pain has now been specified. Location: Conclusions - middle of first paragraph.

Comment: As with chronic fatigue, a small proportion of the patients in the CRIS dataset will be in receipt of psychological therapy to help manage persistent pain, with this as the reason for their having a CRIS record. So I wonder whether a future study using the linkage with the LDN could explore referral patterns for such therapy.

Response: This has been added as future work in the Conclusion section, second paragraph.

Reviewer 2

Comment: The authors have written about how well issues of pain are captured by mental health professionals within their electronic health record. This paper highlights how pain issues may be more or less prevalent or detected by clinicians. The authors cite some relevant literature but I think the framing of the paper and the discussion could be stronger if focused on how these results might inform clinical services and assessment. There is considerable literature on the relationship between pain and serious mental illness that could be better highlighted. It is well established that people with SMI are less likely to report pain than those without an SMI and the findings of this paper support that clinicians are less likely to record that within their mental health record, despite the high likelihood that they are experiencing pain. The implications for these findings for how they could shape clinical care and the need for additional training to ensure that clinicians are probing and recording about pain issues for those with SMI seems an important topic to explore further.

Response: A paragraph to address these comments has been added to the Discussion section - third paragraph.

Comment: The methods seem appropriate with the 80/10/10 approach and the sources of data used to create their dataset with one exception of the inclusion of the LDN dataset. It appears that they used a GP dataset that is not just limited to mental health records. A fuller justification for why they explored differences between the two datasets would be helpful.

Response: More justification for looking at why the two datasets were explored has been added.

Location: Introduction - Background Rationale section - last paragraph.

Comment: A little more detail about the NLP methods would also be helpful as much of that is explained in their prior papers but is not as well described in this paper.

Response: More details have been added about the NLP application within the Methods - Recorded Pain section.

Comment: The use of logistic regression is appropriate, though the rationale for the inclusion of model 4 could be made stronger.

Response: More justification has been on the inclusion of model 4 - in the Results section at the end of the paragraph following Table 2.

Reviewer 3

Comment: On the important aspects that are missing: 1) nowhere does it become clear what the problem is that is being addressed. The background says “the complex and potentially bidirectional nature of this relationship requires further understanding”, but why does it require further understanding? What don't we know? 2) the conclusion claims that the results have significant implications for assessment and management of pain, but those implications are never mentioned. While the results are worth publishing because they show that physical pain is often reported in mental health notes, the study did not distinguish between ‘asked about pain and none reported’, ‘asked about pain but not recorded’, and ‘not asked about pain’, so to claim implications for assessment and management of pain seems premature.

Response: 1) More information around this has been added in the Introduction - Objectives section - second paragraph. 2) Information has been added in the Discussion section (5th paragraph) to address this comment.

Comment: It would be helpful if the authors define what they mean by ‘recorded pain’ early on. I initially assumed it was pain intensity. Please add some example of pain terms.

Response: An explanation has been added early on in the Methods section. Some examples have been included too.

Comment: Abstract: Please avoid CRIS , LDN abbreviations as many readers will not know without explanation.

Response: These abbreviations have been removed.

Comment: Please specify the specific implications rather than saying they exist.

Response: Specifics have been added to the conclusion section within the abstract.

Comment: Strength & Limitations: Please rephrase and look for strength and limitations of the methods. The current list, with exception of the last bullet, is neither strength nor limitation.

Response: This section has been rephrased and updated in the Strengths and Limitations section.

Comment: Background: The second and third sentence are general statements but the citations to support them are specific to depression [1] and to HIV [2]. Please either add different literature that supports the general statements or rephrase.

Response: References [1] and [2] have been updated to different literature. Reference [2], while described as being specific to HIV, the authors state that when developing the framework, they included factors that were important in patients with HIV and in patients with chronic pain independently, which is why it was cited in this research. Location: Introduction - Background Rationale.

Comment: Demographic features don't “influence” pain perception. People with different features may experience pain differently, but the features themselves do not affect pain perception. Please rephrase.

Response: This has been rephrased. Location: Introduction - Background Rationale - third paragraph.

Comment: The paragraph about SES feels disconnected. Please drop or add why it is relevant to this study.

Response: More justification has been added for the inclusion of SES. Location: Introduction - Background Rationale - end of fourth paragraph.

Comment: Please clarify the distinction between primary and secondary care data, as this may not be clear for all readers.

Response: More details have been added to distinguish primary and secondary care data within the Introduction section.

Comment: Methods: Under 'Participants', please clarify what "active (i.e., under accepted referral)" means.

Response: This has been rephrased for clarification.

Comment: Under Recorded Pain: please add examples of pain terms so readers do not have to refer to the lexicon that is described elsewhere. Please also briefly describe the accuracy of the application that is used to classify sentences, so readers do not have to go somewhere else to find out about performance of the application.

Response: Some examples of the pain terms have been added, along with a link to the lexicon as a footnote, for easy access. The F1 score of the NLP application has now been included.

Comment: Please include some examples of 'metaphorical mentions of pain'.

Response: Examples of metaphorical mentions of pain have been added within the Recorded Pain subsection.

Comment: About codes for pain medications and diagnoses it says "While these codes were developed for chronic pain, they are generic enough to be used for this research" Please add how it was determined that they could be used for this study. The authors might want to consider mentioning this a potential limitation of the study.

Response: I have added some examples of the codes that were used, which show they were quite generic and so can be used in this instance.

Comment: 'Another application' was used to determine anatomical location. Please briefly describe who developed the tool, how it works, as well as its accuracy in classifying body parts.

Response: More details have been added about the anatomy application.

Comment: Results: 18188 had pain key words but only 14202 relevant. Please add examples of pain words that were not relevant.

Response: Examples of not relevant pain mentions have been added.

Comment: It would make sense to combine Cohort characteristics and the Pain mentions section.

Response: This has been combined.

Comment: For Table 2 please describe what is being predicted.

Response: This has been added to the table caption.

Comment: Page 12, line 11: "compared to the remainder of the sample" is not correct. Given that these are regression models, the comparison is always with the group that makes up the reference, not the remainder of the sample. Related, on line 21 about 'independent increased odds', this is not correct. The comparison is Asian vs. White and Black vs. White. Not Asian vs. Black. Please rephrase.

Response: Page 12, line 11 has been rephrased. Line 21 has also been rephrased.

Comment: Odds ratios becoming stronger or strengthening is vague. Please rephrase. The authors probably intend to say that the ORs are higher, compared to another model, but since they are not comparing model fit we can not say whether higher is better, more accurate, or stronger etc. On the topic of not comparing model fit, what is the purpose of models 2 and 4? I suggest just reporting unadjusted and model 3.

Response: These sentences have been rephrased and emphasis made on the importance of unadjusted and model 3, with some justification for why models 2 and 4 are included.

Comment: On anatomy distribution, please rephrase “the nature of the pain”. It suggests labels like ‘stinging’, ‘dull’, ‘throbbing’, but what is meant is anatomical location.

Response: This has been rephrased.

Comment: It would be of interest to see the distribution of locations by SMI (yes/no).

Response: Information about the top 3 locations by SMI and non-SMI has been added following Table 3.

Comment: Discussion: Some language in the discussion is inaccurate. For example, females being overrepresented, gender disparities in pain experiences. More female patients had pain mentions. Overrepresented means that there were more female patients than should have been. ‘Disparities’ suggests some external factor that caused the difference in pain mentions. This study was purely descriptive and did not look into those factors. Please rephrase and avoid using those terms.

Response: This section has been rephrased, removing the use of words such as disparities and overrepresentation.

Comment: The suggestion that the results align with undertreatment of minority patients should be removed. As there can be multiple reasons why some groups have higher pain mentions than other (see main concern 2 above) it is premature to suggest this is the result from undertreatment.

Response: This sentence has been removed.

Comment: A limitation of the study is not that it depends on physicians recording, but that you cannot distinguish whether asked and not recorded or not asked at all.

Response: This limitation has been added to the Discussion section, and wording rephrased to indicate the comment.

Comment: The sentence “While the NLP application achieved good performance metrics during its development and evaluation” needs a citation.

Response: Citation for this has been added.

Comment: Generalizability of an exploratory and descriptive study like this is not the point of the study. I suggest dropping the part about generalizability.

Response: This has been deleted.

Comment: Please drop the last paragraph of the discussion. The study was descriptive so cause & effect is not at stake. Moreover, the study did not show disparities. It did show differences in recorded pain mentions but the cause of that was not addressed by this study and hence the authors cannot say that disparities exist.

Response: The last paragraph has been deleted.

Comment: Conclusion: Please omit the last sentence as it cannot be concluded from this study. The study looked at pain mentions in clinical records, not at the relationship between pain and mental health or interventions to manage pain.

Response: The last statement mentions looking at relationship between pain and mental health as well as interventions only as part of future work. We did not intend to mean that the current work could answer this. The line about interventions has been removed, and a statement has been added highlighting the need for longitudinal data in order to look at any relationships.

VERSION 2 – REVIEW

REVIEWER	Kelly , Erin Thomas Jefferson University Hospitals
REVIEW RETURNED	26-Feb-2024

GENERAL COMMENTS	The authors were responsive to my prior comments and the manuscript is improved. There are still a few odd word choices that should be addressed. Abstract: the final sentence of the conclusion is under-developed “Additionally, targeting improved detection for minority and disadvantaged groups can promote health equity.” Targeting what and by whom? Introduction: The descriptions of pain and its association with other variables could be sharper. Are you referring to all pain conditions (chronic and acute) or just symptoms? You reference that there is a complex relationship between mental health and pain but do not provide any examples of what that means. Within your results you talk about how the magnitude of the odds ratios are increasing for some groups as variables are added or removed. However, I would caution that the analytic method does not test whether these increases are significantly changing between models and therefore I would not interpret them as such. They remain significant as other factors are added and removed but the change in magnitude is not being tested. Table 2 footnote: New language is a bit odd “Outcome is recorded or no recorded pain.” Instead, please use something like: Outcome is recorded pain vs no recorded pain. Throughout the paper, it is more common to refer to the joint concepts of race and ethnicity rather than ethnicity alone especially with the categories that the authors used.
--

REVIEWER	Meerwijk, Esther VA Palo Alto Health Care System, Center for Innovation to Implementation (Ci2i)
REVIEW RETURNED	13-Feb-2024

GENERAL COMMENTS	Thank you for addressing my concerns.
---------------------------------------

VERSION 2 – AUTHOR RESPONSE

Reviewer 2

Comment: Abstract: the final sentence of the conclusion is under-developed “Additionally, targeting improved detection for minority and disadvantaged groups can promote health equity.” Targeting what and by whom?

Response: This has been updated to reflect targeting improved detection of pain by care providers.

Comment: Introduction: The descriptions of pain and its association with other variables could be sharper. Are you referring to all pain conditions (chronic and acute) or just symptoms? You reference that there is a complex relationship between mental health and pain but do not provide any examples of what that means.

Response: I am referring to all pain conditions and symptoms, regardless of whether they are acute or chronic. I have tried to clarify this by adding a line in the first paragraph of the Background Rationale. I have also added some examples from literature on the complex relationship between mental health and pain in the second paragraph of the Background Rationale. I hope this provides more clarity.

Comment: Within your results you talk about how the magnitude of the odds ratios are increasing for some groups as variables are added or removed. However, I would caution that the analytic method does not test whether these increases are significantly changing between models and therefore I would not interpret them as such. They remain significant as other factors are added and removed but the change in magnitude is not being tested.

Response: The increase was meant purely as an increase in the actual numbers, but I see how this can also be read as an increase in magnitude. I have removed words such as increased/higher/reduced, to avoid this misinterpretation.

Comment: Table 2 footnote: New language is a bit odd “Outcome is recorded or no recorded pain.” Instead, please use something like: Outcome is recorded pain vs no recorded pain.

Response: This has been updated.

Comment: Throughout the paper, it is more common to refer to the joint concepts of race and ethnicity rather than ethnicity alone especially with the categories that the authors used.

Response: This is a very good point. I have added to the Variables – Demographics section that this is what I mean when stating ethnicity, just for simplicity. However, if you believe that I should mention race and ethnicity in every instance, then I am happy to make that change.